

# Use of aerial thermography to reduce mortality of roe deer fawns before harvest

Jan Cukor[1,2], Jan Bartoška[3], Jan Rohla[1], Jan Sova[4,5] and Antonín Machálek[6]

[1] Faculty of Forestry and Wood Sciences, Czech University of Life Sciences Prague, Prague, Czech Republic
[2] Forestry and Game Management Research Institut, v.v.i., Prague, Czech Republic
[3] Faculty of Economics and Management, Czech University of Life Sciences Prague, Prague, Czech Republic
[4] Workswell s.r.o., Prague, Czech Republic
[5] Faculty of Engineering, Czech University of Life Sciences Prague, Prague, Czech Republic
[6] Research Institute of Agricultural Engineering, p.r.i., Prague, Czech Republic

## ABSTRACT

In agricultural landscape, there are thousands of young wild animals killed every year. Their deaths are caused mostly by agricultural fieldworks during spring harvest. Among the affected animals there are also fawns of roe deer (*Capreolus capreolus*), which react to danger by pressing themselves against the ground in order to be protected from predators. There were various methods tested in the past aimed at decreasing roe deer mortality caused by agriculture machinery with varied levels of success. This contribution presents technology that documents the possibility of searching for fawns with a thermal imaging device carried by an unmanned aerial vehicles (UAV). The results are based on field research that estimated the ideal height of flight being ±40 meters above ground. If the climatic conditions are favourable, it is possible to monitor and mark fawn locations using GPS coordinates in an area of about 14 ha in 25 minutes, which is the average flight time of UAV on one battery charge. The thermo-camera is very reliable in finding fawns in early morning hours (4 to 6 a.m.) when there is the highest temperature contrast between the searched object and its surroundings. The main limiting factors are climatic conditions and the short time span in which the thermo-camera can be used. If the basic requirements are met, the rate of successful fawn detection can be even up to 100%. An undisputed advantage of this method is the possibility of involvement of local gamekeepers. Thus the agricultural fieldworks are not interrupted.

# INTRODUCTION

Over recent decades, growing competition in the agricultural sector has encouraged the development of modern harvesting machines whose high efficiency goes hand in hand with increased speed of travel. Working speed of the machinery exceeds 4.1 m/s and the cutter bars reach 14 m or more in length (*Steen et al., 2012*). Although it is very difficult to assess the extent to which wildlife populations are affected by agricultural operations, we

Corresponding author
Jan Cukor, cukor@fld.czu.cz

are facing a indisputably negative impact of harvesting machinery (*Morris, 2000*). For these reasons, the number of animals killed or injured during routine agricultural operations has increased dramatically (*Kaluzinski, 1982*).

During fodder crop harvest, agricultural machinery negatively affects a vast number of animal species (*Steen et al., 2012*). Of the class Mammalia, European hare (*Lepus europaeus*) and roe deer (*Capreolus capreolus*) are most threatened by the agricultural operations described above, as juveniles in danger press against the ground trying to protect themselves from predators (*Steen et al., 2012*). Roe deer fawns are "hiders" and during the first 4–6 weeks mostly stay hidden alone in the vegetation (*Jarnemo & Liberg, 2005*) and their mother visits them 3–7 times/day (*Espmark, 1969*). For this reason, there is a high risk of young roe deer being killed by the cutter bars. Moreover, adults are also threatened as they often prefer grassland habitats, even in comparison to other agricultural localities (*Linnell, Nilsen & Andersen, 2004*). The season of fodder crops harvest in Central Europe also overlaps with the roe deer birth season (*Linnell, Wahlström & Gaillard, 1998*; *Moorten et al., 2008*) whose peak falls between 20th May and 10th June (*Linnell, Wahlström & Gaillard, 1998*).

Accurate numbers of game killed this way were documented only in a few studies. In Sweden, roe deer fawn mortality due to fodder crops mowing was assessed to be within 25–44%. The results were based on a three-year study between years 1997–1999 (*Jarnemo, 2002*). Along with the red fox (*Vulpes vulpes*) predation, farming activities are presented as the main cause of roe deer fawns mortality (*Jarnemo, 2002*; *Jarnemo et al., 2004*). In West Germany, the number of killed roe deer fawns was estimated at 84,000 individuals in a single year 1979 (*Kittler, 1979*). It is important to realize that these data come from a time when the mowers speed was substantially lower than now and the cutter bars were shorter.

Since an increase in roe deer populations is mainly driven by a growing number of juveniles, lowering fawn mortality is essential to maintain the roe deer population stable or even increase it. Various approaches are adopted to reduce game mortality during fodder crops harvest. For instance, in the past, plastic bags, installed in the area of risk on the day before harvest, were used to scare away the game (*Jarnemo, 2002*). Losses can be significantly mitigated by modifications in harvesting technology, e.g., by mowing from the centre of the field section (*Frawley & Best, 1991*; *Humbert et al., 2010a*; *Humbert et al., 2010b*), leaving higher stubble or reducing the speed (*Steen et al., 2012*). Another way to tackle the problem may be to leave a portion of the field to be harvested later (*Frawley & Best, 1991*). To reduce the mortality of larger animals, thermal imaging cameras mounted on the mowers were successfully tested (*Steen et al., 2012*). However, the protection of detected game has to be carried out by farmers, which results in considerable delays in agricultural operations. Therefore, for the time being, the use of mounted thermal imaging cameras has not spread significantly.

An alternative to the use of thermal imaging cameras that would not directly restrict the flow of harvest can be thermal imaging cameras carried on unmanned aerial vehicles (UAV). First tests of thermal imaging cameras mounted on UAV were carried out in 2014 (*Christiansen et al., 2014*) but the image frames failed to reach required quality. This method of searching for game and its transfer to safe locations is designed to be carried out

by gamekeepers, which is advantageous for farmers as it does not interfere with the flow of agricultural operations.

However, the use of unmanned aerial vehicles is restricted by a number of country-specific safety and legislative measures. The present article documents possible ways to apply UAV equipped with thermal imaging cameras to be used for searching roe deer fawns in the harvest season when the young fawns are most vulnerable. Our approach develops and innovates the methods described in the trial operations in Denmark in 2012 (*Steen et al., 2012*) and 2014 (*Christiansen et al., 2014*). The aim of the article is to describe and analyse the prevention-research activities of UAV equipped with a thermal imaging camera. A partial aim of the study was to find out the time consumption of the used method and possible limiting factors of UAV utilization.

# MATERIALS AND METHODS

## Study area

Experimental UAV flights equipped with thermal imaging cameras took place from 17 May 2016 to 17 June 2018 in 10 selected localities in the Czech Republic (Fig. 1). Most flights were carried out over sites managed by the Czech University of Life Sciences Prague in Farm Estate Lány (Lány-Ploskov, WGS: N50°5.87017′, E13°48.89828′; Lány-Amálie, WGS: N50°6.38994′, E13°51.34900′). Other sites were selected in agreement with agricultural subjects that were willing to provide their agricultural land for research activities. All selected localities were managed for standard fodder production of alfalfa (*Medicago sativa* L.). Detail information of acreage of each field is mentioned in the 'Results' section.

The premises of the University Farm Estate Lány belonging to the Czech University of Life Sciences Prague were selected for their sufficient area of alfalfa, which is very attractive crop for roe deer to hide in and feed on. Harvested alfalfa was 40–70 cm high, the thick cover was difficult to walk through and search for the game (Fig. 2).

## Equipment

For the experimental UAV thermography search for roe deer fawns, the research team used Workswell WIRIS 2nd generation thermal camera mounted and carried on a HEXACOPTER GD HX-1100F ZODIAC UAV. DJI GO application run on an iPad mini was used to visualize the data from the thermal imaging camera. Flights were planned by DJI Ground Station App run on an iPad mini as well. Images taken with the Workswell WIRIS were evaluated by WorkswellCorePlayer software.

## Thermal imaging camera and its setting

Game was monitored by LWIR (long-wave infrared, i.e., wavelength of 7–14 m) Workswell WIRIS [2nd] gen thermal imaging camera manufactured by the Czech manufacturer Workswell, s. r. o. The selected thermal camera had lens of focal length $f = 13$ mm (corresponding to FOV 45° × 37°) and micro bolometer resolution of 640 × 512 pixels. The camera was able to record surface temperatures in the range of $-25$ °C to $+550$ °C. Accuracy of the recorded temperature stated by the manufacturer is $\pm 2$ °C or 2% of the range (the worse of the data applies to the given measurement); thermal sensitivity of the

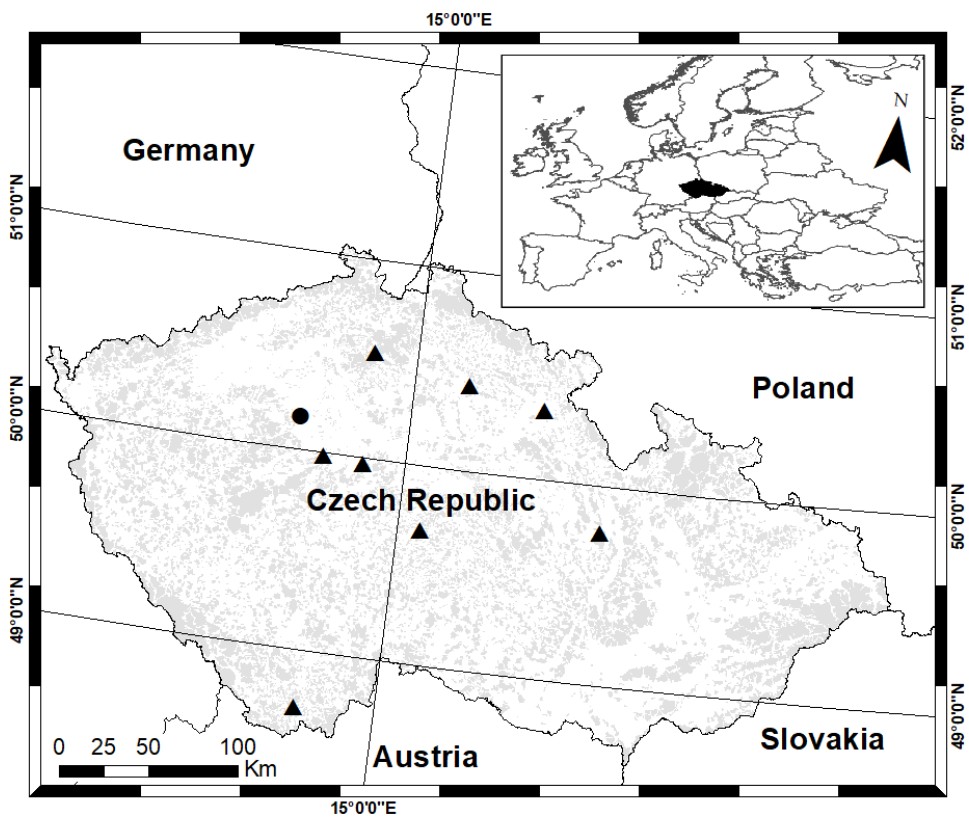

**Figure 1** **Localization of the main study area Lány (●) and other study sites (▲).** Grey coloured area shows distribution of forests in the Czech Republic.

camera is ≤50 mK at 30 °C. The total weight of the thermal camera is approximately 400 grams. The camera has 32 GB built-in memory and HDMI for video output. Video can be transmitted in real time to the ground station via wireless technology, which was used in our case. A correct UAV speed of flight over the monitored site was important. For our purposes, the ideal flight speed ranged from 4 to 6 m/s (4.6 m/s in average). A higher speed limits the user's ability to scan the images. Due to the limited battery capacity, slower flights would significantly reduce the area that can be scanned. Imaging frequency of the UAV thermal cameras ranges from 9 Hz to 30 Hz. For our purposes, 9 Hz was sufficient.

A key feature of WIRIS is also the ability to connect with GPS technology and accurately store the GPS position of a localized object. Therefore, it is possible to monitor and mark the locations of the roe deer fawns by GPS coordinates under suitable conditions. The flight trajectory can be planned ahead using e.g., DJI Ground Station App (Fig. 3). The application allows estimating the expected flight time, which is very important in relation to limited battery life.

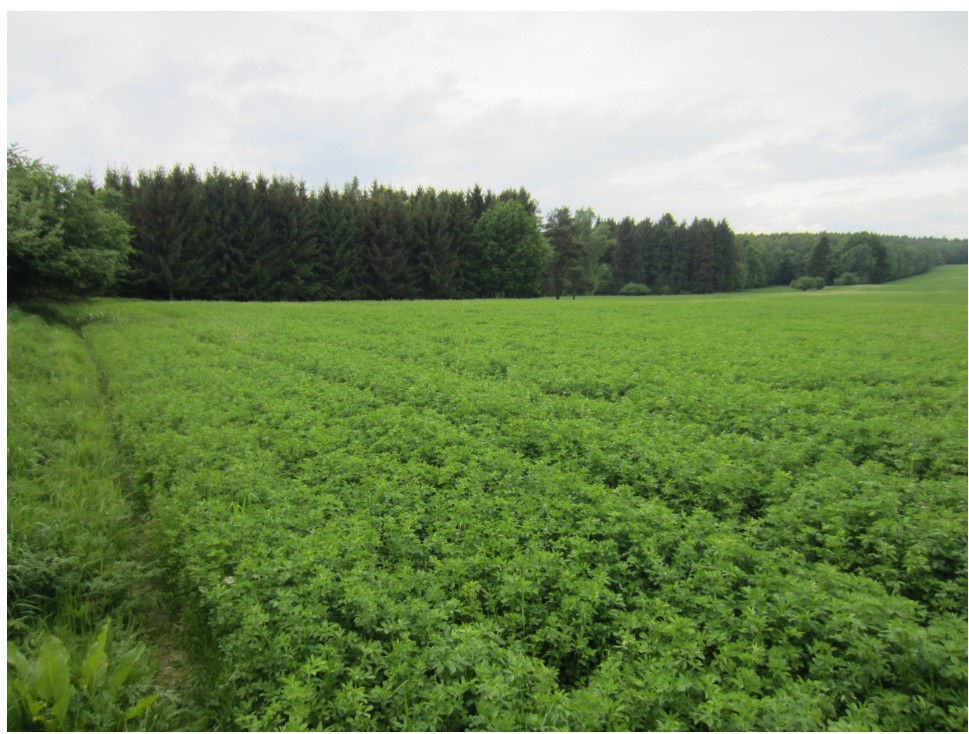

**Figure 2** **Typical alfalfa field used for the search of hidden roe deer fawns (study area Lány, 26th May 2016).** Photograph by Jan Bartoška.

## Setting thermal imaging camera parameters

As a measuring instrument, the thermal imaging camera does not measure temperature directly but calculates it on the basis of the radiation intensity of the measured surface and specified parameters.

From a metrological point of view, the standard ISO 18434-1:2008 distinguishes qualitative and quantitative thermography. The quantitative thermography works with real temperature values and, for these purposes, correct emissivity values, reflected apparent temperature, and parameters correcting atmospheric influence (i.e., atmospheric temperatures, relative atmospheric humidity, and the distance between the subjects). In the qualitative measurement we do not set the parameters and we are able to analyse the situation on the basis of colour changes on the thermogram, corresponding to the changes in the scanned intensity of the heat radiation. But we either have to ignore the surface temperature information or consider it the "apparent temperature".

From a methodological point of view, our measurement is qualitative, and we do not necessarily need information about the surface temperature, as we are only interested in the location of the sought-after game. However, approximate information about the surface temperature of the scanned objects may be useful, with regard to the effective adjustment of the temperature scale mode, as discussed below. For this purpose, the parameters can be set as follows: emissivity is set to 0.95, which is approximately the value that corresponds to the emissivity of the soil, vegetation and game fur. We set the reflected apparent

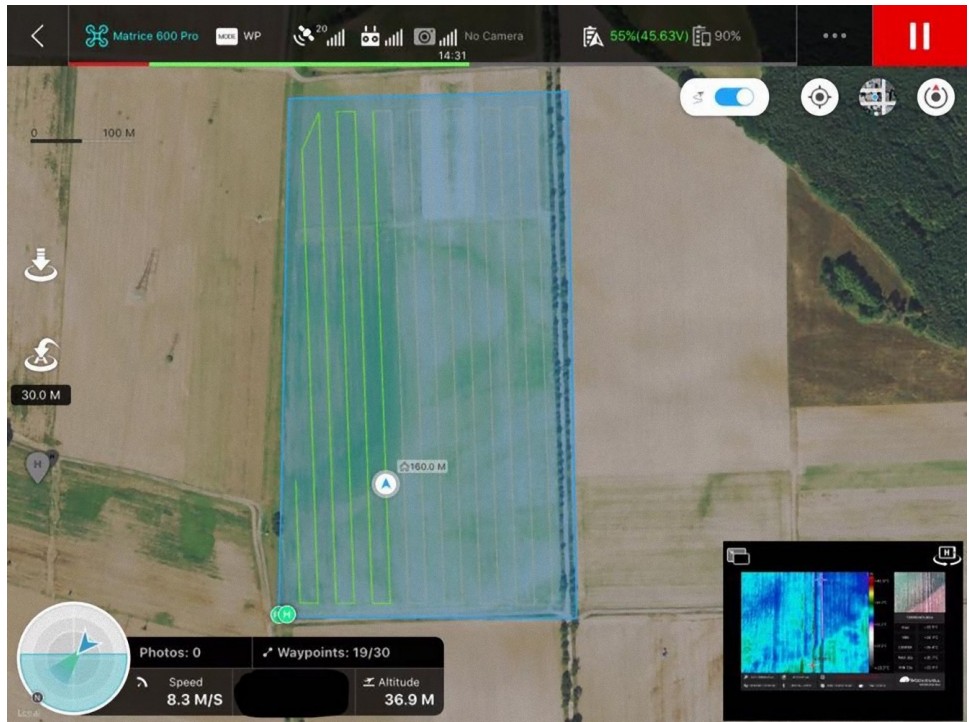

**Figure 3  Example of UAV DJI M600Pro flight plan programmed in DJI Ground Station App— illustrative image.** Workswell WIRIS 2nd gen is compatible with the software. Therefore, the flight plan could be watched at once.

temperature by directing the camera, with emissivity set to 1, to the atmosphere (the so-called direct method, see ISO 18434-1:2008). The average value the camera will show in an area corresponding to a part or the whole image is then set as the reflected apparent temperature for a given measurement. The process can be simplified by setting the value to $-40\,°C$ for clear sky and atmospheric temperature when the sky is overcast. It should be pointed out that this parameter does not play a crucial role because of the high emissivity of the scanned surfaces. That is why the setting can only be approximate. We also set the atmospheric parameters very roughly, as the effect on the assessment of the temperature is even more negligible in our case. In view of the setting, the measurements should be referred to as apparent temperature, although this figure may not differ fundamentally from the actual temperature.

## Suitable palette choice

Thermographic data are displayed in a thermogram in a colour palette, which matches colours with apparent and actual temperatures in accordance with a predefined colour scheme (Fig. 4). It is therefore necessary to select a suitable palette before each use of the thermal camera. The sepia or rainbow palette, or their modifications, proved to be the most suitable for our purpose. These palettes provide sufficient colour contrast even at small temperature differences. However, it should be emphasized that the choice is a matter of

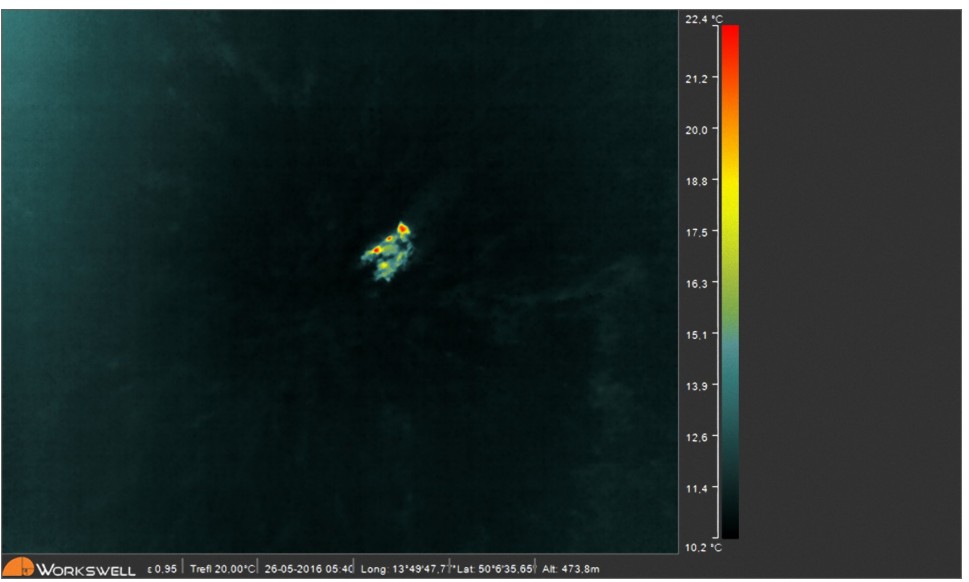

**Figure 4   Two roe deer fawns captured on a thermogram.** Rainbow palette was used for visualisation of apparent temperatures.

individual preference. It depends on the pilot and the UAV imaging system which palette will eventually be chosen as the most effective for the given application.

## Temperature scale choice

Today it is standard in thermography that the temperature scale of the thermal camera can be operated in two modes: automatic and manual. In the manual mode the upper and lower scale limits have to be set manually. This requires knowledge of appropriate values. In the automatic mode, the upper and lower scale limits are changed so that the temperature scale is set to display the warmest and the coldest areas in the thermogram. When setting the temperature scale to the automatic mode, the upper and lower scale limits automatically change according to the temperature changes in the scanned scene. Values of actual or apparent temperature then correspond to different colours from the colour palette at different moments. This might be a problem as constant colour changes reduce the attention of the pilot when locating the fawns, especially if it takes some time. As a result, the fawn may be overshadowed and its subsequent death follows.

Setting the scale to the manual mode therefore proved more suitable for our purposes. However, the upper and lower limits must be set correctly. During the flight, these limits can be regulated; for example, if the whole scanning takes longer and the range needs to be adjusted, e.g., because of the warming of the soil caused by the rising sun etc.

An optimal setting of the upper and lower scale limits can be found, for example, after spotting the first fawn. The surface temperature of the animal and the background (vegetation, soil) is then set. Limit values must be set with certain tolerance and, overall, the whole setting must be convenient for the UAV pilot.

## Temperature homogeneity of the thermogram

Since the surface temperature of the game we want to locate may not show a substantial temperature difference from the surrounding objects (the typical temperature difference ranges from 5 to 10 °C but may be even smaller), it may be important to ensure a sufficient temperature homogeneity in the thermogram. This can be achieved by the NUC (non-uniformity correction) and thermal stabilization of the thermal imaging camera before a flight (*Vollmer & Möllmann, 2010*). NUC is a process in which the transfer characteristics of each micro bolometer within the micro bolometer field of the thermal camera are mutually unified. This leads to a significant reduction of the temperature inhomogeneity of the image and overall noise.

Temperature stabilization of the thermal imaging camera usually takes 7 to 15 min. During the process, the camera is switched on and the internal electronics are heated; the measured temperature and homogeneity of the thermogram may change in the meantime. If the lens get fogged, it is necessary to wipe the lens with a soft cloth, as even a thin coating of water is virtually non-transparent for infrared radiation in the LWIR band.

Figure 5 shows the temperature profile function demonstrating the inhomogeneity of the thermogram before NUC (Fig. 5A) and the result obtained after NUC (Fig. 5B). A homogeneous surface with surface temperature of approximately 20 °C was chosen as the scanned object to show the effect of correction after NUC as well as the thermal imbalance in the original thermogram (*Minkina & Dudzik, 2009*). Inhomogeneity in the thermogram before NUC (A) and after performing NUC (B) was represented by a temperature profile function from one corner of the thermogram to another. It is the thermogram of a surface which is almost temperature-homogeneous. Before NUC, the difference between the maximum (34.5 °C) and the minimum (30.5 °C) temperature is about 4 °C from one corner to another, but in practice it can be significantly greater (Fig. 5). After NUC, inhomogeneity dropped to about 1 °C, but in practice the value may even be smaller. By reducing the inhomogeneity in the image, the detection capabilities will significantly increase and the risk of missing the object will decrease. Thermogram homogeneity of the Workswell WIRIS is significantly higher than the homogeneity of FLIR ONE 3, which significantly simplifies object search. This means that search with Workswell WIRIS can be done faster and the probability of overriding the search object is lower. However, it is necessary to expect a higher price and worse UAV operability used for this thermal imaging camera.

## Maximum flight altitude setting

In order to reliably detect the presence of a particular object, the camera lens must project the observed object to a detector area that corresponds to a real area of at least $3 \times 3$ pixels of a Focal Plane Array (FPA) micro bolometer. The following formulas Eqs. (1) and (2) can be used for the calculation of a horizontal and vertical field of view. There are separate formulas for the vertical and horizontal dimension as is customary in the branch. We need to ascertain $h$ maximum horizontal and $v$ maximum vertical dimension of the object, which will be fully displayed on the thermal imaging camera detector, i.e., Eq. (1)

$$h = 2d\tan\left(\frac{hfov}{2}\right) \tag{1}$$

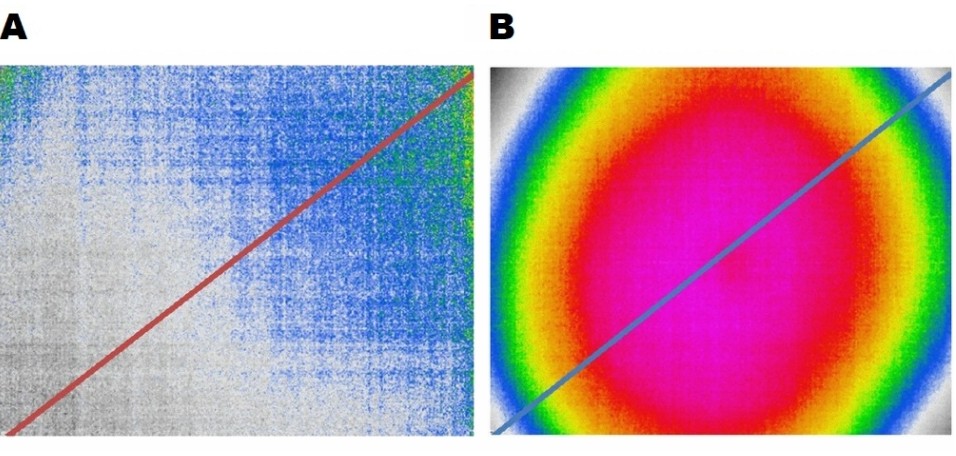

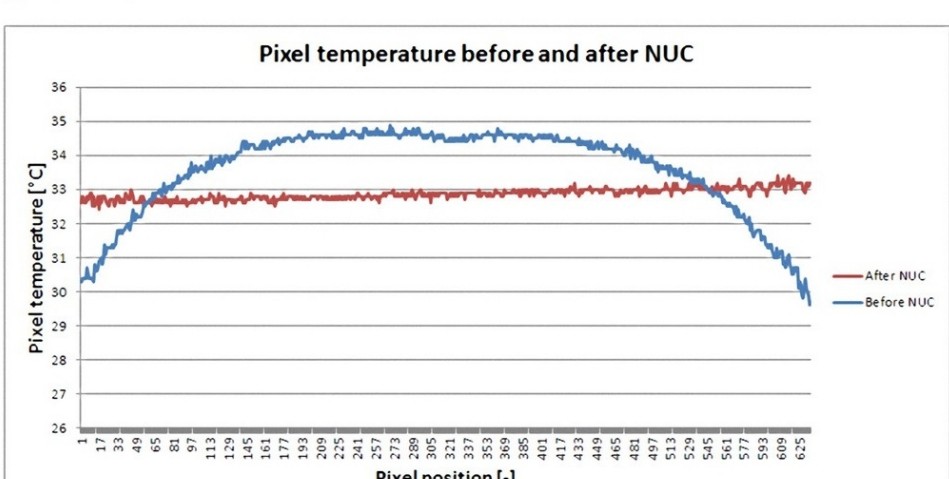

**Figure 5** **Inhomogeneity in the thermogram after NUC (A) and before NUC (B) is represented by the temperature profile function from one corner of the thermogram to the other; the graph on the bottom (C) shows inhomogeneity of the thermogram before and after NUC using the measurement function line profile.** On the $X$ axis, the sequence number of the pixel on the black line is entered. On the $Y$ axis, the temperature that corresponds to that pixel is shown. It is evident that after the correct execution of the NUC procedure, the inhomogeneity of the thermogram will significantly decrease and the probability of finding the subject will thus increase.

resp. Eq. (2)

$$v = 2d \tan\left(\frac{vfov}{2}\right) \tag{2}$$

where *hfov* is the horizontal field of vision and *vfov* is the vertical field of vision indicated at the lens of the camera, *d* is the distance of the camera and the scanned object for which *h* and *v* values are determined. To get a pixel value, we need to divide the *h* and *v* by the corresponding number of horizontal and vertical pixels.

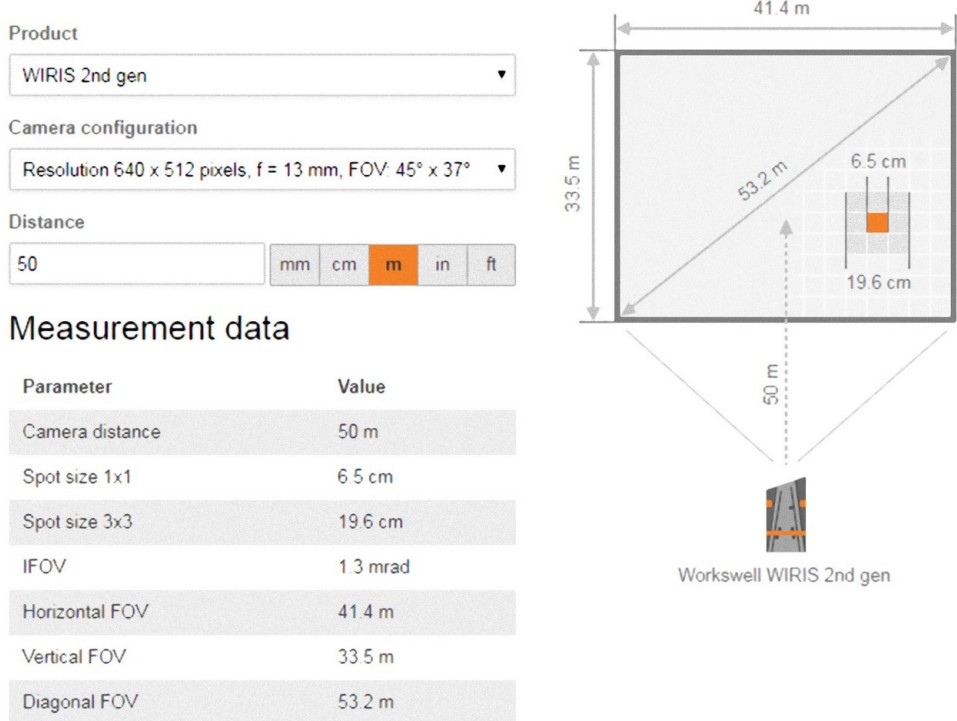

**Figure 6** **Calculation of the field of vision and the smallest detectable object in the Workswell WIRIS 2nd gene thermal imaging camera, depending on the type of lens selected and the flight altitude.** In our case, 13 mm (hfov = 45°, vfov = 37°) lens and flight altitude of 50 m was decided for. The smallest reliably observable object has dimensions of approximately 20 cm × 20 cm. For that calculation, Eqs. (1) and (2) were used.

If we assume the size of a fawn to be e.g., 20 cm × 20 cm, the formulas Eqs. (1) and (2) show us that for lens of $f = 13$ mm (hfov = 45°, vfov = 37°) and camera resolution of 640 × 512 pixels, the maximum flight altitude would be approx. 50 m. See Fig. 6.

## Applied UAV

To search for game using the Workswell WIRIS 2nd gen camera, we used a HEXACOPTER GD HX-1100F ZODIAC UAV with a maximum take-off weight of 16 kg. The maximum weight of the cargo in a stabilized gimbal can be 4 kg. Such load allows the maximum flight time of 25 min on one battery set. This UAV can be operated manually or semi-automatically by an autopilot implemented in a UAV's control unit. Manual operation enables the operator to gradually cross the field and immediately locate the game and let the UAV hover over the spot. Semi-automatic control with autopilot allows setting the flight path before the flight and then, during the flight and return, it gives feedback to a pilot in case a hot spot (which could be the searched for game) appears. A completely automatic UAV flight is not allowed for legislative purposes in most countries and the UAV pilot must always be ready to take over control of the UAV whenever the situation requires.

A commercially available DJI M600Pro (Fig. 7) type UAV was also tested for carrying the thermal imaging camera. The weight of this UAV with the WIRIS Thermal Camera

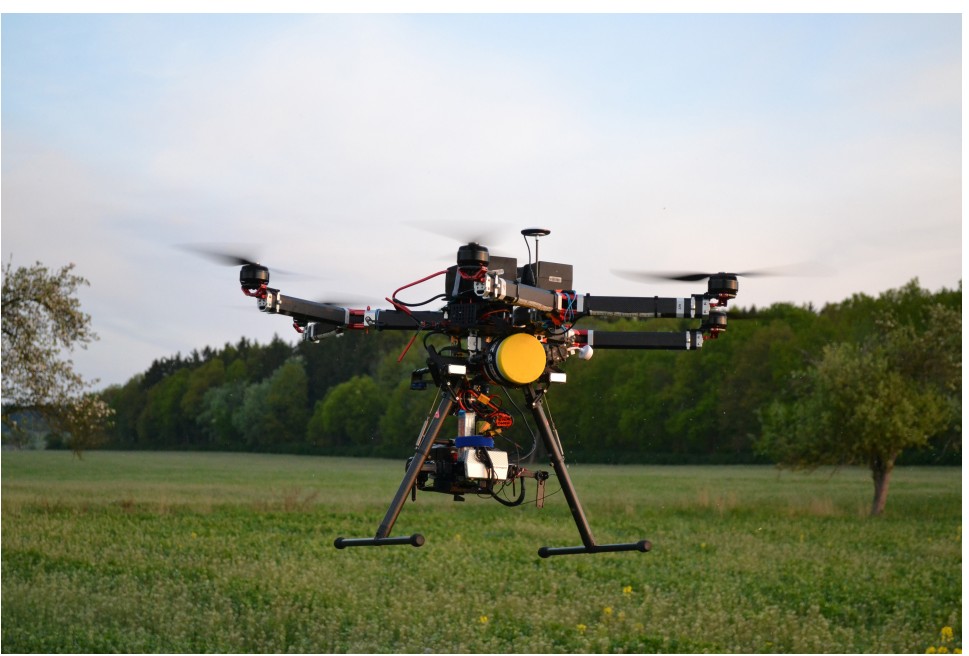

**Figure 7** **HEXACOPTER GD HX-1100F ZODIAC UAV equipped with a Workswell WIRIW 2nd gen. camera.**

was approximately 10 kg, the flight time was about 25 to 30 min at a maximum speed of 18 m/s. Image transfer from the thermal camera was provided by DJI Lightbridge 2, to be consequently displayed in DJI GO and DJI Ground Station apps, which also enable the operator to plan the entire flight trajectory in advance. DJI M600Pro proves this method can be applied using commonly available UAVs.

Both tested UAVs, i.e., the DJI M600 Pro and the HEXACOPTER GD HX-1100F ZODIAC UAV, have proven to be equal in terms of utilization. The only practical difference was the fact that the DJI M600Pro had a few minutes longer flight time on average. However, the DJI M600Pro represents a significantly lower cost of ownership, with the whole set being (without the thermal camera) taken for approximately 11,000 USD (excluding VAT). In all cases, a Workswell WIRIS Thermal Camera (10,000 USD excluding VAT) was used. Therefore, the total price of the report was around 21,000 USD at the time of writing. However, in the foreseeable future, we can expect a significant price cut of one third or more.

## Data collection

The UAV operation falls within the civil aviation category and is therefore subject to binding legislative rules (Civil Aviation Act No. 49/97 (*Ministry of the Interior of the Czech Republic, 1997*) and its Implementing Decree 108/97 Coll.). In the Czech Republic, UAV are certified and surveyed, and pilots are licensed by the Civil Aviation Authority (CAA).

The search for the game animals using the thermal camera took place in the selected locations from 5 to 7 am. Morning hours were chosen because of the highest possible

temperature contrast between the sought-after game and their surroundings. The duration of aerial scanning was limited by an increasing sunlight intensity which quickly reduced the differences between temperatures of the game and their surroundings. The weather thus directly influenced the applicability of the technology.

Under the law, unmanned aerial vehicles can only be used under favourable climatic conditions. For unmanned systems, Appendix X of the L2 Regulation states that flights can only be conducted at a distance limited to the direct visual contact of the pilot with the unmanned device (it is 200 to 500 m without using visual aids), so that the pilot can correctly evaluate the visibility, obstacles and also the surrounding air traffic. This technology is further limited by morning fog, in which case the flights and subsequent harvest had to be cancelled. A UAV flight may only be conducted in such a way as not to jeopardize the safety of air traffic, persons and property on the ground and the environment. UAV flights cannot be performed everywhere. In general, flights are banned over densely populated areas (cities, municipalities), over people without their consent, over communications, railways, or near airports. Aircraft companies may have an exemption from these rules.

The UAV flights over selected alfalfa fields were always performed from the edge of the monitored field block. The UAV flew up to the required altitude (see section Testing), and the alfalfa field was then systematically monitored until the battery capacity was depleted. If the battery power supply dropped below 20%, the UAV returned to the start position for safety reasons. This value is not fixed and can be set as needed. After replacing the batteries, the monitoring of the selected field block continued until the site of the field block intended for subsequent mowing was completely scanned.

Images from the thermal camera were transmitted in real-time to a display placed on a tripod near the point where the UAV took off for the check-out. When an object with a significantly higher temperature was detected, GPS coordinates of the place with the expected occurrence of the game were stored. After crossing over the entire monitored field block, the GPS coordinates were subjected to a field survey and the fawns were carried to the nearest safe location by the gamekeepers. Tufts of grass (alfalfa) were used for lifting the fawns to avoid undesirable contact with human skin.

### Fodder crops harvest technology

The harvest always took place in the morning (from 7 to 8 a.m.), immediately following the camera-equipped UAV flights. The timing of the operations was mainly opted for to eliminate the potential risk of the fawns returning to the fodder field from which they had been brought to a safe place.

Fodder crops were harvested in the locality Lány by a John Deere tractor (JD 7930, 164 kW) fitted with a front-mounted and side-mounted disc mower, the mower being driven by the PTO shaft of the tractor. The total length of the cutter bars was 6.1 m; the average speed of travel was 4.2 m/s. Other localities were harvested by commonly used type of tractors equipped with cutter bars. The harvesting method was the same in all localities. The alfalfa field was divided into smaller parts, which were subsequently cut from the centre to the edge, i.e., towards safe areas (neighbouring fields or forest stands).

**Table 1  Characteristics of test flights.**

| Flight altitude (m) | Scanned area (m) | Pixels on the item | Number of test flights | Successful detection |
|---|---|---|---|---|
| 25 | 21 × 17 | 9 × 12 | 2 | 2 |
| 50 | 42 × 34 | 4 × 6 | 2 | 2 |
| 70 | 56 × 47 | 3 × 4 | 5 | 5 |
| 100 | 84 × 67 | 2 × 3 | 5 | 5 |
| 120 | 101 × 81 | 2 × 2 | 10 | 9 |

**Notes.**

Flight altitude, UAV height above ground; scanned area, field of view of used camera given selected altitude; pixels on the item, the size of test object on image in pixels; number of test flights, number of test flights repetitions; successful detection, number of cases the test object was detected (from number of test flights) given selected altitude.

# RESULTS

## Testing

First, test UAV flights were conducted to determine the optimum flight altitude (Table 1) in the second half of May 2016. Five-litre plastic containers were randomly placed in the high-grown alfalfa field of acreage about 5 hectares as test items. The containers had an approximate size of 30 cm × 40 cm × 20 cm. The plastic containers were filled with water of 17 to 25 °C, which corresponds to the approximate temperature of a fawn's surface. The test flights were performed in the early morning (5–7 a.m.) as well as the real search for roe deer fawns. The whole field was standardly explored by the pilot of UAV according to the flight plan (see Fig. 3). The position of containers was not known to the pilot. For each test flight only one plastic container was placed in the field. The decisive criterion was the successful detection of the test items and the size of the scanned area.

The number of test flights was increasing steadily with the flight altitude of UAVs in order to achieve as accurate assessment of the successful attempts as possible. Within an altitude range of 25 to 100 metres above the terrain, the object's detection on the display by the UAV operator was always 100%. At a flight altitude of 120 metres, the detection rate was only 90% because of a low number of displayed pixels on the tested item in the flight altitude. Therefore, it is possible to indicate an altitude of flight between 50 and 70 m as an ideal height for the highest detection of plastic containers or roe deer fawns in practice. Scanned area for flight altitudes 50 m and 70 m was 1,428 m$^2$ and 2,632 m$^2$, respectively.

## Practical utilization in the field

The results of roe deer fawn detection using a UAV-carried thermal imaging camera are processed in Table 2. The table shows that the flight altitude of ±35 m using Workswell WIRIS thermal camera above the terrain was most commonly used. Average flight duration within whole 13 flights was 1.76 min per hectare (±0.29 SD), i.e., about 13.9 hectares per capacity of one battery set which is about 25 min. In the flight altitude of 35 m the flight duration was slightly higher (1.87 ± 0.15 SD), which is caused by a smaller acreage of the scanned area (see Table 1).

The observed efficiency of roe deer fawn detection was up to 100% search success rate. After the harvest, each particular field block was walked through, but no killed fawns were

**Table 2  Practical utilization of UAV and the results of detection of roe deer fawns in the Czech Republic during 2016–2018.**

| Date | Study site | Area of the field (ha) | Flight altitude (m) | Flight duration (min) | Number of detected fawns | Number of killed fawns |
|------|-----------|-----------------------|--------------------|-----------------------|--------------------------|------------------------|
| 17/05/2016 | Srbsko | 4.1 | 35 | 8.2 | 1 | 0 |
| 26/05/2016 | Lány–Amálie | 9.6 | 40 | 15.1 | 2 | 0 |
| 02/06/2016 | Lány–Ploskov | 7.7 | 35 | 13.7 | 1 | 0 |
| 08/06/2016 | Pohledy | 13.2 | 35 | 22.1 | 1 | 0 |
| 22/05/2017 | Lány–Amálie | 9.6 | 70 | 9.4 | 2 | 0 |
| 26/05/2017 | Horní Krupá | 7.3 | 30 | 14.9 | 1 | 0 |
| 30/05/2017 | Lány–Ploskov | 7.7 | 35 | 13.7 | 1 | 0 |
| 09/06/2017 | Blatce | 4.0 | 35 | 8.2 | 1 | 0 |
| 25/05/2018 | Kacálkova Lhota | 24.3 | 40 | 35.3 | 2 | 0 |
| 09/06/2018 | Buš | 7.2 | 40 | 11.9 | 1 | 0 |
| 10/06/2018 | Nahořany | 13.8 | 30 | 26.4 | 2 | 0 |
| 11/06/2018 | Lány–Ploskov | 8.4 | 30 | 16.8 | 2 | 0 |
| 17/06/2018 | Hrudkov | 13.2 | 30 | 25.5 | 7 | 0 |

found in any of the cases described in our article. During the flights, UAVs were also able to find and rouse adult deer, which fled to a safe neighbouring area. However, adult game was not monitored any further. The detected roe deer fawns were carried to a safe site by the gamekeepers, i.e., to the nearest woods or another field not intended for harvesting.

## DISCUSSION

This paper documents a feasible utilization of a UAV-carried thermal imaging camera to search for roe deer fawns before fodder crops harvest. The issue had not been addressed properly so far. Fawns killed by agricultural machinery were originally mentioned by *Kaluzinski (1982)*, but not in actual relation to their possible protection; only for summarization of game losses. The first article about roe deer fawn protection was published more recently (*Jarnemo, 2002*). The tested method was based on the utilization of black plastic sacks on 2 m long poles to scare female roe deer from placing their fawns in the field prepared for harvest and thus make the females remove the ones already hiding in the place. This method shows a relatively high efficiency. Of 22 fawns bedded in the vicinity of sacks, 18 were moved by their mother on the day when the sacks were set out and three fawns were moved by their mothers on the next day. The distribution of one sack per hectare (as in *Jarnemo, 2002*), requires a considerable effort especially in larger fields of alfalfa or meadows of several dozens of hectares. The indisputable advantage of the plastic sack method is the simplicity of utilization and low cost of its realization. Another possibility is to search for roe deer fawns by gamekeepers with trained hunting dogs and volunteers. This method, however, is associated with the risk of overseeing the fawn in the meadow (*Cukor et al., 2018*).

Modern thermal-imaging-camera technology was first used for animal detection in 2012 to detect chickens and rabbits like experimental animals (*Steen et al., 2012*). The cameras were mounted on the mowers; the success rate of the method described was

100%. Yet this technology has not gained sufficient support, perhaps due to the fact that it delayed harvesting operations. In case an animal was detected, the farm staff would have to interrupt mowing and transfer the animal to a safer place. The article published in 2014 (*Christiansen et al., 2014*) describes the first tests of thermal imaging cameras mounted on UAV also on chickens and rabbits. However, vibrations inside UAV and strong wind caused poor-quality images and, therefore, UAV technology was dropped. Consequently, thermal cameras were mounted on high-lift trucks, which ensured a precise altitude setting and high-quality image. The best results in this case were found for an altitude of 3–10 metres; at an altitude of 20 metres the results of detection and recognition of animals were less accurate. However, the mentioned articles described only possible detection of experimental animals in defined (ideal) conditions and not in the practice.

Our primary aim was to detect the roe deer fawns whose life was threatened by mowers. The flights were conducted at an altitude of approximately 35 meters; in the 'Results' section we confirmed up to 100% success rate in roe deer fawn detection, i.e., the same rate of success as in the case of thermal imaging camera carried on mowers (*Steen et al., 2012*). The technical equipment allowed scanning tens of hectares in several tens of minutes. The presented solution offers an undeniable advantage in fluency in agricultural work, which is not compromised by our method. UAV flights are conducted in early morning hours and are planned to be completely organized and performed by gamekeepers. Farmers would therefore be able to start mowing with a minimal risk of killing deer.

Climatic conditions are in fact the main factor that might threaten or thwart the UAV scanning. Thermal imaging cameras carried on UAV are mostly protected by IP 50 certification (*National Electrical Manufacturers Association, 2004*) which excludes utilization in rainy weather. Usage in hazy weather is not limited from the point of thermal camera damage; however, the detection rate is reduced. It is therefore possible to ensure a successful detection of roe deer fawns by a lower flight altitude. On the other hand, in the rainy weather the fodder harvest is mostly cancelled and the UAV monitoring as well as the harvest can be performed within next days. For the safety flight it is necessary to reflect a wind speed. In general, it is recommended to fly when the wind speed is lower than 4.4 m/s (*Junda, Greene & Bird, 2015*). Another limiting factor of UAV utilization in practice could be the acreage of field blocks. The area of field blocks has increased steadily in the past and the average field size in the Czech Republic in range of 21–30 hectares is one of the highest in the European Union (*Figala, Prchalova & Tester, 2001*; *Reuter & Eden, 2008*). The number of scanned hectares can be increased by using additional battery sets.

Moreover, it is possible to increase the efficiency of UAV utilization via targeted selection of habitats which are preferred, e.g., bedding sides of roe deer fawns. Recent research shows that about sixty percent of all roe deer fawns in grassland are located no further than 50 m from the edge of a field (*Christen, Janko & Rehnus, 2018*). This knowledge allows searching for fawns in a larger acreage of meadows or fields with alfalfa. A UAV operator in cooperation with hunting management personnel (hunters, gamekeepers and game managers) could search for fawns at the edge of the field where the probability of fawn presence is higher.

Another aim of UAV utilization should be the detection of juveniles of smaller animal species such as European hare (*Lepus europaeus*) or nests of grey partridge (*Perdix perdix*) and other animals. Thermal sensitivity and resolution of thermal imaging cameras carried by UAV is increasing. Therefore, the issue might be addressed in the near future.

## CONCLUSION

In the present paper, a method of using thermal imaging cameras carried by UAV in research and prevention is described in detail. The method is expected to significantly reduce mortality of roe deer fawns. The article reports on the first described functional use of a thermal imaging camera carried by UAV validated both in test and real operation in the Czech Republic. Field flights confirmed up to 100% success rate of roe deer fawn detection on the scanned field. As for practical utilization, the effectiveness could be influenced mostly by human factor (loss of attention, tiredness). Also, the fundamental conditions for successful detection were described, i.e., optimum flight altitude of $\pm 40$ metres above the terrain, early morning hours with sufficient temperature contrast, and appropriate resolution of the camera.

An undisputed advantage lies in the fact that the method can be applied and performed by gamekeepers before the fodder crops harvest. Therefore, it causes no interruption in the flow of agricultural operations, often precisely planned in relation to climatic conditions and ripeness of the crops.

The challenge for the future use of UAVs and carried thermal cameras is to search for other species of small game and other animals that are more difficult to detect than larger species such as roe deer.

## ACKNOWLEDGEMENTS

The authors thank all gamekeepers and farmers who hve provided the study sites for the field monitoring. We would also like to thank Petr Lněnička who realized the fieldworks (UAV monitoring). We are also grateful to academic editor and three anonymous reviewers for their benefitial comments and suggestions that greatly improved our manuscript.

### Funding

The paper was written within the frame of project NAZV provided by Department of Agriculture Czech Republic—Program of Complex sustainable systems in agriculture 2012–2018, Funder Id: 10.13039/501100006533, QJ1530348. Study was also partly financed by the Internal Grant Agency of Faculty of Forestry and Wood Sciences, CULS Prague (IGA B03/18). There was no additional external funding received for this study. The funders had no role in study design, data collection and analysis, decision to publish, or preparation of the manuscript.

## Grant Disclosures

The following grant information was disclosed by the authors:

Department of Agriculture Czech Republic—Program of Complex sustainable systems in agriculture 2012–2018: QJ1530348.

Internal Grant Agency of Faculty of Forestry and Wood Sciences, CULS Prague (IGA B03/18).

## Competing Interests

Jan Sova is an employee of the Workswell s.r.o. The authors declare there are no competing interests.

## Author Contributions

- Jan Cukor conceived and designed the experiments, performed the experiments, prepared figures and/or tables, authored or reviewed drafts of the paper, approved the final draft.
- Jan Bartoška conceived and designed the experiments, performed the experiments, analyzed the data, authored or reviewed drafts of the paper, approved the final draft.
- Jan Rohla authored or reviewed drafts of the paper, approved the final draft.
- Jan Sova analyzed the data, contributed reagents/materials/analysis tools, authored or reviewed drafts of the paper, approved the final draft.
- Antonín Machálek authored or reviewed drafts of the paper, approved the final draft.

## Data Availability

The raw data is available in a Supplemental File.

## Supplemental Information

Supplemental information for this article can be found online at http://dx.doi.org/10.7717/peerj.6923#supplemental-information.

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
