# Peer review of "Use of aerial thermography to reduce mortality of roe deer fawns before harvest"

_PeerJ, doi:10.7717/peerj.6923_

## Round 0.1 · original submission · Minor Revisions

The reviewers have found the paper generally well written. There are several minor comments that should be addressed. It is recommended to shorten the introduction (streamline the paper). It is also suggested to improve the quality of graphics.

Reviewer 1 ·

Basic reporting

The whole text is clear, generally well written and interesting.
Figures and photos are useful, well chosen.
The reference paragraph needs some small editing.

Experimental design

It is well described and documented. I would suggest adding the costs of the equipment

Validity of the findings

The whole device seems really effective and can reduce the fawn mortality in the cultivated fields

Additional comments

The paper is generally well written and interesting. The terms of the problem and the experimental design are well described. The devices can actually reduce fawn mortality. I would suggest adding an estimate of costs of the equipment (drone plus thermal imagine camera). The reference paragraph should be better edited. I have here included some suggestions to improve the text in specific points.

Suggested corrections/notes for specific points:

row 50: fawns of roe deer
row 77: (Blodgett et al., 1995; Gardiner, 2006)
84: grey partridge
86-87: juveniles in danger press against the ground
87: Roe deer fawns are “hiders” and during the first 4-6 weeks mostly stay hidden alone in the vegetation.
90: as they often prefer
92-93 please cancel from “62%” onward. This is the birth timing for a specific population studied in Sweden. Please add a more generic sentence with a more general birth peak (for ex. 20th May-10th June). On the argument see (and quote) Linnell, Wahlström, Gaillard 1998 “From birth to independence: birth, growth, neonatal mortality, hiding behaviour and dispersal”, in Andersen, Duncan, Linnell, eds. “The European roe deer: the biology of success”. Scandinavian University Press (this is currently the best review).
94 (but also 134, 331, 427): game animals/species
98: in (West) Germany
101: and roe deer population size was quite lower than now
102: Please add something like “Since roe deer increase is mainly driven by the recruitment of juveniles, lowering fawn mortality is essential to maintain stable or increase the population”.
147: alfalfa or lucern
412: This paper documents
416: was published more recently (Jarnemo et al., 2002)
453: personnel (hunters, gamekeepers, game managers)
475: Meloidae
477: please put the year of publication in the correct place
479 (also 479, 513, 542): please use small letters in the title
481-482 (also 494): scientific name in italics
482: Viltrevy, Swedish Wildlife 1969; 6
489: breeding bird

Reviewer 2 ·

Basic reporting

Style and language are carefully chosen and used terminology is appropriate at most places. However, the manuscript does not always read easy. Most importantly, many sentences seem too long and somewhat complicated. Wording is confusing in some places and there are recurrent repetitions of the text throughout different sections of the manuscript (e.g. line 107 and 417-421). Second, the paper structuring, though formally mostly correct, is somewhat cumbersome. The presented work describes deals largely with two things. It aims at validating the use of thermal imaging cameras mounted on unmanned aerial vehicles (UAV) to detect middle-size mammalian species such as the roe deer fawns in the agricultural land. It also aims at demonstrating in field the applicability of the method to detect roe deer fawns in the conditions of central European agriculture with the goal of aiding prevention of machinery related fawn mortality. To conclude, the manuscript would benefit greatly from having a professional English speaker reading and editing the text in a way that would respect the natural logic of the paper that goes from 1) scientific validation of the method, 2) technical guidance for the reader with some account of the affordability for the hunting clubs, 3) example of applying the method in the wild (i.e. practical example of roe deer fawn detection on a few fields in the Czech Republic).

The account of the context for the presented study in the Introduction is adequate as is the use of literature. Some inconsistency in the format of citations both in the text of all sections and in the References shall be corrected. The last part of the Introduction shall give a clearer idea of the research question. Currently, the only lead is the sentence on the line 125-128 which is too vague. Is the aim of the study to validate the use of thermal imaging mounted on UAV for detecting middle size mammals? How was the validation performed? Given the method is proved valid for the purpose, i.e. the detection rates are reasonably high, is the next aim to show the applicability of the method in the natural settings including some technical guidance on using the tool for the reader?

Figure and legends. Visual quality of figures is excellent and their relevance adequate. Labelling is generally insufficient. Figure legends shall be fully explanatory for all variables displayed. Some of my concerns in detail below:
Figure 1. Consider displaying Lány premises by one type of symbol and the rest of study places by a different symbol.
Figure 2. Complement the figure legend with the place and date of the picture. E.g. “Typical alfalfa field used for searching roe deer fawns. The site is from this and this place, from this and this time.”
Figure 3. “Example of a plan…”, and put nd in superscript (2nd)
Figure 5. What is the graph on the bottom? It is not explained in the figure legend.
Figure 6. This looks more like a software interface when setting the parameters of the camera. Strictly speaking, it is not a “calculation” which would be represented by sort of formula/equation. Rather it is an “example” of how you may set up the parameters in a study like yours. To me it feels like the primary aim of this figure is to illustrate that the software interface is user friendly and easy to deal with (even for hunters who may not be proficient in using such a type of device). Only after you convince a reader (hunter/hunter manager) that the method is easy to utilize and affordable, you can get it in practice.

Raw data were not supplied.

Experimental design

The primary research seems to fit the Scope of the Peer J journal.
The research question is at present not developed sufficiently and shall be elaborated on in more detail, particularly at the end of the Introduction (see my comment above).
The investigation was performed to a high technical and ethical standard.
Methods behind using and setting the cameras and UAVs are described with sufficient detail. However, the field procedures lack enough information to allow replication of this study. Specifically, the following shall be explained: how many plastic containers were placed in the grass at every time (line 383)? How did you choose the location of the containers in the field? Randomly? Did the pilot of the UAV know the location of the containers he/she was searching for? How many replications were performed? From table 1 it looks like the number of replications may have been 2-10 which is not too many. When did you perform the flights? Some information on the time of the day is provide on line 332 but I can not recall seeing any information on the part of the year. How exactly did you determine the optimal speed being 4-6 m/s (line 358)? Did you evaluate detection rates of fawns at different speeds and found this to have highest detection rate? Why did you use different sizes of scanned areas (Table 1)? If the aim was to determine optimal flight height/altitude in an experimental approach, one has to keep everything same in the replication except manipulating the factor of interest (here flight height, or perhaps flight speed). If one does manipulate the two factors (here flight height and area) how can you be sure the optimal height is not in some way confounded by changes in the scanned area? During the validation, you can derive detection rate of containers as you know their location but how did you conclude the detection rate is 100% in searching for live fawns? How can you be sure that you did not miss any fawn by the camera? I assume there is still a chance that you miss a fawn, it just happens to survive the mowing and so you do not find its corpse.

Validity of the findings

The results seem pretty strong. However, there are possible flaws in the experimental design that need some addressing from the authors (please see my comments above). Also, I am not sure you can really state you have 100% detection rate. OK, perhaps you can get close to 100% but not exactly 100%. How can you assure the camera/software or whatever of the technicalities never get in a problem? Given there is always a human element, how can you be sure you do not overlook a fawn during the flight? These may be too anxious concerns, but authors shall provide rigorous arguments for a reader not to have any of such.

Additional comments

Below please find some minor comments concerning mostly style, language and typing errors.

Majority, if not all, of citations refer to agricultural practice in Europe. Agricultural policies in Europe are different from other areas, e.g. USA (http://www.momagri.org/UK/points-of-view/A-comparative-approach-to-European-and-American-agricultural-policies_798.html
or
http://www.europarl.europa.eu/RegData/etudes/ATAG/2016/586615/EPRS_ATA(2016)586615_EN.pdf ). For this reason, I encourage the authors to make clear early in the text that the paper is dealing with a problem mostly relevant for the European agricultural practice.

Citations in the text: inconsistent use of commas and semicolons in the text citations. Comma is missing, f. ex., on line 77 in „Blodgett et al. 1995“, when it shall be „Blodgett et al., 1995“. The citation on line 77 is „Blodgett et al. 1995, Gardiner; 2006“, while it shall be „Blodgett et al., 1995; Gardiner, 2006“. Similarly line 79, 81, 82, 91, and elsewhere in the text)

Abstract
Line 55: UAV not AUV

Introduction
Line 69-71 and 74-75: The two sentences are very similar and so repeating them is redundant. Please consider merging the two sentences.
Line 72: Could you please provide some numbers that would illustrate how dramatic the increase in mortality related to agricultural operations may be during the years?
Line 76: Maybe better to directly state which taxonomic groups (beetles, Orthoptera,..?) are concerned.
Line 77: “in birds” better than “class Aves”
Line 82: According to BirdLife Int (http://datazone.birdlife.org/home) the English name of Crex crex is Corncrake, Perdix perdix is Grey Partridge.
Line 84: Can you please provide any reference here?
Line 93: Is the 25th of May and 7th of June a typical fodder harvest season? If so, where? CZ? How about annual variation?
Line 94-101: please use litter and fawns. Please be consistent in using terms litter and fawn throughout the manuscript
Line 96: please delete “Altogether with”
Line 96: Better provide the exact years of study and add citation. E.g. “…to be within 25-44% during the years XX-YY (citation)”
Line 98-99: Consider rephrasing to make the text shorter. E.g. “In Germany, the number of killed roe deer fawns was estimated at 84 000 individuals in a single year 1997 (Kittler, 1979)”
Line 106: consider moving the sentence “In the past,… (Jarnemo et al., 2002)” to the line 102, after the sentence “…crops harvest.” In that way you would start with what was done in the past and follow with more recent approaches including the thermal imaging cameras as the most state of the art approach.
Line 110: Please consider deleting the part of the sentence “using such…process. In this case,” in order to omit the redundant text. The proposed shorter version may sounds, e. g., as following: “However, the protection of detected game…operations.”
Line 113: is the increased labor demanded by the farmers the only reason? What are the financial costs/benefits of using the cameras/UAVs for the farmers/hunters?

Materials and Methods
Consider moving text from the line 130-135 to the last paragraph of the Introduction
Line 133-134: This is repetition of line 85-90 and can be omitted.

Study area
Line 138: better “…in selected 9 locations in the Czech Republic, from which XX site are experimental sites of the Czech University of Life Science Farm Estate Lány (Fig. 1)” Please state the size of the study areas, e. g. by referring to table 2 (please note that you need to refer to it as table 1 if it’s the first table to be referred to). What did the study sites look like? All fields for fodder production? Please see the comment to line 391 for inconsistency in the number of study sites.
Line 139: it seems there is something wrong with the coordinates. E 13° 48’?
Line 139: abbreviation CULS shall be explained here. It only is explained later, on line 146.
Line 141-142. The last sentence “Selected…in Fig. 1” is redundant and can be omitted.
Line 146-149: better to merge the two sentences as stating both is redundant. E.g.: Leave the first sentence “…to hide in and feed on” and delete the sentence “Alfalfa is not only…landscape”.

Equipment
Line 157,167, 308, 312, 329 and elsewhere: “2nd” (nd in superscript)
Line 161-163. I am bit confused. Cheaper solution than what? So, you used two types of thermal cameras? You mention lower resolution in one type. Did you in any way test/analyze the effect of lower resolution on detection rates of the roe deer fawns?

Thermal imaging camera and its setting
Line 172: at instead of @
Line 183: I am sorry, I do not understand. Do you say that you monitor area of 20 ha in 15 min. by one flight? What is the average UAV flight time? How did you manage to fly for longer than is average flying time? Did you save batteries in some way?

Setting of thermal imaging camera parameters
Line 201-204: the sentence is long and reads difficult. Consider shortening, e.g. by adding brackets “…influence (i.e. atmospheric temperature, relative atmospheric humidity), and …” and by deleting “are required” at the end of the sentence.
Line 215: any citation available here?
Line 234: “given” or “required” seems better than “this”
Line 241-243: I am confused. There is no account about the manual mode in the paragraph. Should the sentence on line 241 read as “In the manual mode, the upper and lower…”?
Line 246: “time periods” better than “moments”?
Line 246: problem for what? For detectability rates?
Line 252: “optimal” better than “ideal”?

Temperature homogeneity of thermogram
Line 261: NUC is not defined here, it is defined only later in the text, on line 268. Please define NUC here.
Line 280: “…significantly greater (Fig 5c?)”
Line 281: better “smaller” than “lesser”
Line 286: How much lower is the probability of non-detection? This and similar issues about detectability rates shall be elaborated on in the text as those seem critical aspects of the utility of the described method.

Maximum flight altitude setting
Line 294: the sentence seems awkward. Please reformulate. Also, it seems there shall be a citation instead of “[32]” at the end of the sentence.

Applied UAV
Line 313: there is a different hexacopter type stated in the legend of the figure 7. Please correct.
Line 316: what is the difference in detection rates between manual and computer control?

Data collection
Line 331: move the first sentence to line 335 after “.. . 108/97 Coll.).” and before “Morning…”
Line 332: for the sake of consistency (see e.g. line 367), please use “4 to 6 am” instead of “4 to 6 o clock in the morning”
Line 339: Do you mean “applicability” rather than “availabilitity”?
Line 352: There is no “section 3.1” anywhere in the manuscript
Line 358 and 373: better be consistent and either use m/s or km/h throughout the manuscript.

Fodder crops harvest technology
Line 371: Was JD7930 used at all sites?

Results
Line 388: Avoid vague statements of the type “this and that characteristics is given in table 1.” (see also line 400). Be specific. Describe the detection rates here, how these varied as a function of the flight height (and other characteristics).

Table 1.
Line 391: “Results” of the test flights performed at 13 sites in the Czech Republic during 2016-2018.

Discussion
Line 417: delete “which was also described in this publication”
The work by Jarnemo et al 2002 is already mentioned in the Introduction (line 106-108). Why to repeat it here?
Line 420: “as” better than “when “,
Line 420: “…of bag 18 were moved by their mother on the day when the sacks were set out and 3 fawns were moved by their mother on the next day”
Line 422: Remove the sentence “Plastic …spacing” and put it in the following sentence, e.g. “Distribution of one plastic bag per hectare (as in Jarnemo et al. 2002), requires….., which are common in present agricultural landscape in the Czech Republic (Figala et al. 2001).”
Line 425: “advantage of the plastic bag method…” rather than easy-to-confuse “described method”.
Line 427: what game species were detected?
Line 428: remove “In this case,”
Line 431: “safer” better than “safe”;
Line 431: Please add the species of Christiansens study: “The article on this and that species published in 2014…”
Line 435: remove “flight” since the sentence refers to high lift trucks and not UAV?
Line 445. Remove “previously prepared meadows and fields” and consider changing “with minimal risk of killing” for “without the risk of killing”
Line 447. Adverse or rainy weather also stops mowing, so the negative effect of bad weather on the UAV applicability does not seem as much a big problem to me.

Conclusion
Line 460: remove “submitted”
Line 436: “carrying” better than “carried”

References
Some references use capital letters (line 513, 533, 542) while the rest not. Please correct.
Line 477: year of publication placed wrong in the reference
Line 481,494,511: species name shall be in italics
Line 486: Grey partridge not gray
Line 531: structure of what? Missing word?

Table 1: all the variables shall be explained. What does “number of test flights” means?
Table 2, legend: “…detection of roe deer fawns at 13 locations in the Czech Republic during 2016-2018”

Reviewer 3 ·

Basic reporting

This is an interesting contribution dealing with an authentic wildlife management problem in agricultural landscapes. Limited literature (at least official peer reviewed literature) is available on this item. English is clear and professional (...but I'm nor English native myself). Abstract is OK. Literature is ok as well as the provided context. Goals are clear.

Neverteless, I perceived some weaknesses, as follows:
1) Introduction is a bit too long, I suggest Authors to delete or shorten some parts (eg, delete from lines 75 to 84);
2) Tests flights methodology should appear in M&M;
3) Results are somewhat uncomplete. In particular, I would have expected data i) on the time needed to exaustively cover the prospected fields (at least raw data, average time/prospected hectare, standard deviation); ii) on the amount of days (during the study period) in which climatic conditions were deemed unfavourable to prospection; iii) on the number of detected adult deer;
4) I'd appreciate a (short) comment on the expected survival of fawns which were removed from alfalfa fields;
5) Discussion on the limiting factors should be more in deep;
6) Figures are too many and not all are of sufficient quality or interest. I suggest to delete Figg. 1 and 8, improve (if possible) the quality of Figg. 3-5-6, and improve the appeal and self standing character of all figures' and tables' captions. In addition, it should be clearer how Authors may state that spots on Fig. 4 are suggestive of two fawns.

Experimental design

No major comment adding to Section 1. Basic reporting

Validity of the findings

No major comment adding to Section 1. Basic reporting

Additional comments

No comment

---

## Round 0.2 · Minor Revisions

Your article proposes a solution of an important problem of co-existence of agriculture/farmers and wild-life. Although it was tested to detect juvenile deer, there is definitely a promise to detect and save other species that may feed on/near crops.

The reviewers have identified several remaining typos and unexplained results (e.g. why it was decided that there were 2 deer in Fig. 4, and not 1?). I hope that it will not take long to make this changes.

Reviewer 1 ·

Basic reporting

I am totally satisfied with the corrections made.
The article merits to be published.

Experimental design

I already gave my positive opinion on it.

Validity of the findings

The results are very interesting and promising.

Additional comments

I am totally satisfied with this last version of the paper. There are only very few typographical errors in the quoted literature.

row 518: scientific Latin name in italics
row 548: space among words
551: space between words
556: Scandinavian

Reviewer 2 ·

Basic reporting

Authors considered all my comments and improved the manuscript considerably.

The language is professional and the text is largely fluent to read.

Experimental design

All my concerns were appropriately addressed.

Validity of the findings

All my concerns were appropriately addressed.

Additional comments

I only have few minor comments.

line 10: Workswell s.r.o. (a space needs to be added between Workswell and s.r.o.). Also is the "m" correctly placed there?

line 85: Sentence "it is very...this way" is a repetition of line 69-71 and my opinion is that it shall be omitted.

line 93: please add "populations" (Since an increase in roe deer populations is mainly...)

line 136: Please move "medicago sativa L." to line 130, where there is the first mention of alfalfa in the text.

Line 137: Sentence "However, alfalfa...deer fawns" is a repetition of line 82-84 and shall be omitted.

Line 214: I believe that "purpose" is correct, rather than "purposes"

Line 258 and Fig. 5. I believe it makes sense to switch Fig. 5A and 5B as reader may be interested first in the situation before NUC and then after NUC to see the improvement.

Line 324: I believe "application" shall be deleted.

Line 327-329: please be consistent and use either "$" or "USD"

Line 337 ("The search... 5 to 7 a.m.") and line 339-343 ("Morning hours...technology") shall better fit either on line 354 or 361.

Line 404: Perhaps shorter: "Scanned area for flight altitudes 50 m and 70 m was 1428 m2 and 2632 m2, respectively"

line 410: delete "about". uncertainty is given by provided SD

Line 438 and 475-476: Perhaps shorter to join the sentences in one, e.g. "The area of field blocks have increased steadily in the past to reach on average 21-30 hectares in the Czech Republic in more recent time (Figala et al 2001, Reuter and Eden, 2008)"

Line 477: "by using additional battery sets"

line 495. "100 %" rather than "100%"

Line 497-499: Incomplete sentence?

line 505: " ...that are more difficult to detect than larger species such as roe deer."

line 518. Capreolus capreolus shall be in italics
line 551 "Capreolus capreolus" not "Capreoluscapreolus"

Reviewer 3 ·

Basic reporting

English substantially improved. Literature references and background OK. Structure OK. Raw data (while few) can be visualized. Match results/hypotheses OK. Still, in Fig. 4 it is not explained on which basis Authors can say that those spots are two fawns. Please, be authentically informative on this particular detail

Experimental design

OK

Validity of the findings

OK

---

## Round 0.3 · accepted · Accept

Thank you very much for fast and careful response to all suggestions and comments raised by the reviewers. I hope that your approach will help to protect lives of wild animals and will become a stepping stone in the road to sustainable and eco-friendly agriculture.

#